# On Human-Aligned Risk Minimization

**Liu Leqi**
Carnegie Mellon University
Pittsburgh, PA 15213
leqil@cs.cmu.edu

**Adarsh Prasad**
Carnegie Mellon University
Pittsburgh, PA 15213
adarshp@cs.cmu.edu

**Pradeep Ravikumar**
Carnegie Mellon Universit
Pittsburgh, PA 15213
pradeepr@cs.cmu.edu

## Abstract

The statistical decision theoretic foundations of modern machine learning have largely focused on the minimization of the expectation of some loss function for a given task. However, seminal results in behavioral economics have shown that human decision-making is based on different risk measures than the expectation of any given loss function. In this paper, we pose the following simple question: in contrast to minimizing expected loss, could we minimize a better human-aligned risk measure? While this might not seem natural at first glance, we analyze the properties of such a revised risk measure, and surprisingly show that it might also better align with additional desiderata like fairness that have attracted considerable recent attention. We focus in particular on a class of human-aligned risk measures inspired by cumulative prospect theory. We empirically study these risk measures, and demonstrate their improved performance on desiderata such as fairness, in contrast to the traditional workhorse of expected loss minimization.

## 1 Introduction

The decision-theoretic foundations of modern machine learning models have largely focused on estimating model parameters that minimize the expectation of some loss function. This ensures that the resulting model have high average case performance, which loosely is what is meant by good generalization performance. However, as ML models are increasingly deployed in broader societal settings, and in particular, to assist humans in decision-making, it is clear that humans want models to have not just good average performance but also properties like fairness. Due to the importance of these additional desiderata, there have been a burgeoning interest in capturing these properties via appropriate constraints and modifications of the classical objective of expected loss [8, 12, 18, 27]. In this work, we posit a very natural if simple solution to addressing these varied desiderata that are driven in large part by human considerations. Specifically, we suggest that in contrast to using the standard workhorse of expected loss, we draw from theories of human cognition in psychology and behavioral economics, to consider a "human-aligned" risk instead.

Alternatives to expected loss based risk measures have a long history in decision-making [16], with earlier efforts focusing on percentile risk criteria [11]. In machine learning, instead of minimizing expected loss, various risk measures have been considered in different settings. In risk sensitive reinforcement learning, conditional value-at-risk (CVaR), a percentile risk measure that quantifies the tail performance of a model, has been connected to robustness to modeling errors [5, 21]. Recently, human-aligned risk measures have also been explored in bandit [14] and reinforcement learning [23], where the goal of the agent is to produce long term returns aligned with the preferences of one or more humans.

**Contributions.** In this work, we introduce a novel notion of human risk minimization (Section 3), by bringing ideas from cumulative prospect theory (Section 2) into supervised learning. We explore various salient characteristics of our objective such as diminishing sensitivity, decision-making

based on higher-order moments and information-content or "surprisal" view point of human risk (Section 4). We also study the implications of minimizing our objective in the context of subpopulation performance (Section 5). In particular, our empirical results illustrate that human risk minimization inherently avoids drastic losses across all subgroups.

## 2 Background

### 2.1 Cumulative Prospect Theory

As a seminal work in behavioral economics, cumulative prospect theory (CPT) [28] provides a framework to emulate human decision-making under uncertainty. In particular, CPT points out that humans overweight extreme events that occur with low probability, rather than treating all the events equally, which is the assumption of expected utility theory (EUT). As an alternative to EUT, CPT has three important components [25]:

- Outcomes are considered as gains or losses compared to a reference point;

- Value functions are concave for gains, convex for losses and flatter for gains than for losses;

- An inverse S-shaped (first concave then convex) probability weighting function (Figure 1 (a)) is used to transform the cumulative distribution function so that small probabilities are inflated and large probabilities are deflated [30].

Current machine learning follows the EUT framework in the sense that expected losses are minimized. However, as CPT has pointed out, a human evaluates risk differently. For example, given two models $\{\mathcal{M}_1, \mathcal{M}_2\}$ such that $\mathcal{M}_1$ has zero loss with probability .95 and loses 100 with probability .05 while $\mathcal{M}_2$ loses 5.01 all the time, EUT will choose $\mathcal{M}_1$. The reasoning behind is that the expected loss of $\mathcal{M}_1$ is 5, which is smaller than the expected loss of $\mathcal{M}_2$, which is 5.01. However, from the CPT perspective, $\mathcal{M}_2$ will be chosen because CPT inflates the probability .05 and $\mathcal{M}_1$ will end up having a larger risk. In this case, because of the human-innate probability weighting, we end up choosing a model that avoids drastic losses instead of the one with a better average performance.

The inverse S-shaped CPT probability weighting function captures that humans over-weight extreme events with low probability while under-weight "average" events that are more probable but less extreme. Many parametric forms of the probability weighting function have been proposed [24, 25, 28, 30]. To start with, we formally define a class of weighting functions called $\mathcal{W}_{\text{CPT}}$.

**Definition 1.** *Let* $w : [0, 1] \rightarrow [0, 1]$ *be a differentiable function. Then,* $w \in \mathcal{W}_{CPT}$ *if and only if*

1. $w(0) = 0$ *and* $w(1) = 1$;

2. *there exists* $a \in (0, 1)$ *such that* $w(a) = a$;

3. $w'(x)$ *is monotonically decreasing on* $x \in [0, a)$ *and* $w'(x)$ *is monotonically increasing on* $x \in (a, 1]$.

Traditional CPT probability weighting functions fall into this class, including the original weighting function (for losses) $w(x) = \frac{x^{.69}}{(x^{.69} + (1-x)^{.69})^{1/.69}}$ [28]. For a real-valued continuous random variable $X$ with cumulative distribution function $F(x)$ and a CPT probability weighting function $w \in \mathcal{W}_{\text{CPT}}$, the CPT subjective utility is defined as [25, 28]:

$$U_{\text{CPT}}(X) = \int_{-\infty}^{+\infty} v(x) dg(F(x)) \tag{1}$$

where (1) $v : \mathbb{R} \rightarrow \mathbb{R}$ is a value function; (2) $g(F(x)) = w(F(x))$ when $x < 0$ and $g(F(x)) = -w(1 - F(x))$ when $x \geq 0$.

**Rank-dependent Utility.** As pointed out in [28], CPT subjective utility is a rank-dependent utility since the decision weight on $x$ depends on the "rank" of $x$, which is given by $F(x)$. When $F(x)$ is weighted by $w \in \mathcal{W}_{\text{CPT}}$ [7] and $v(x) = x$, the CPT-weighted rank-dependent utility is:

$$U_{\text{CPT-RD}}(X) = \int_{-\infty}^{+\infty} x \, dw(F(x)). \tag{2}$$

We focus on studying the effect of using CPT-weighted cumulative distribution function $w(F(x))$ on training an ML model. Hence, analyzing the effect of using a reference point and a value function $v$ is out of the scope of this paper. As one may have noticed, if $w(F(x)) = F(x)$, then $U_{\text{CPT-RD}}(X) = \mathbb{E}[X]$.

To have a finite CPT subjective utility [25] for a real-valued continuous random variable $X$ with $w \in \mathcal{W}_{\text{CPT}}$, it is sufficient to ensure $w$ to be strictly increasing on $[0, 1]$ and continuously differentiable on $[0, 1]$, i.e. $w'(0)$ and $w'(1)$ are finite. As proposed by [25], the simplest polynomial that satisfies the above conditions is

$$w_{\text{POLY}}(F(x)) = \frac{3 - 3b}{a^2 - a + 1} \left( F(x)^3 - (a + 1)F(x)^2 + aF(x) \right) + F(x) \tag{3}$$

where $a \in (0, 1)$ is the fixed point, i.e. $w_{\text{POLY}}(a) = a$, and $b \in (0, 1)$ controls the curvature of $w_{\text{POLY}}(\cdot) \in \mathcal{W}_{\text{CPT}}$. As $b$ approaches 1, $w_{\text{POLY}}(F(x))$ will converge to the linear function $w(F(x)) = F(x)$. One could interpret $b$ as controlling the sensitivity of the probability weighting function to a unit difference in probability changes [13]. We use $w_{\text{POLY}}(\,\cdot\,; a, b)$ to denote the polynomial form CPT probability weighting function with fixed point $a$ and curvature $b$.

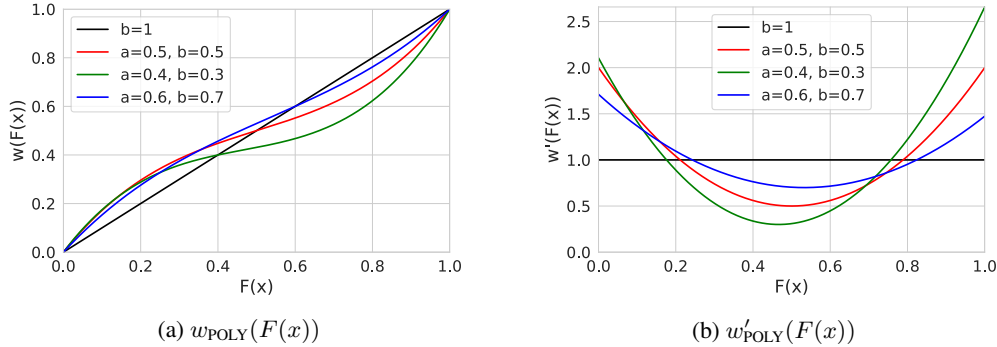

(a) $w_{\text{POLY}}(F(x))$　　　　　　　　　　(b) $w'_{\text{POLY}}(F(x))$

Figure 1: (a) Inverse S-shaped probability weighting function $w_{\text{POLY}}$ is the steepest near the endpoints $0, 1$. The parametric form of $w_{\text{POLY}}(\,\cdot\,; a, b)$ is shown in Equation 3. (b) $U$-shaped CPT probability weighting function derivative $w'_{\text{POLY}}$ up-weights the tails of the original distribution.

**Proposition 1.** *Given any cumulative distribution function $F(x)$, if a non-decreasing continuous function $w : [0, 1] \to [0, 1]$ satisfies $w(0) = 0$ and $w(1) = 1$, then $w(F(x))$ is a cumulative distribution function of some random variable.*

For any $w \in \mathcal{W}_{\text{CPT}}$ that is non-decreasing (e.g. $w_{\text{POLY}}$), for a real-valued continuous random variable $X$ with cumulative distribution function $F(x)$ and density $f(x)$, one can think of $f(x)w'(F(x))$ as a CPT-weighted density and $w(F(x))$ is the corresponding CPT-weighted cumulative distribution function. $U_{\text{CPT-RD}}$ is the expectation of the random variable that has the CPT-weighted cumulative distribution function. We denote the set of non-decreasing functions in $\mathcal{W}_{\text{CPT}}$ to be $\overline{\mathcal{W}}_{\text{CPT}}$.

## 2.2 Empirical Risk Minimization

The canonical way of learning an ML model is through empirical risk minimization (ERM). Given $n$ i.i.d. samples $Z_1, \ldots, Z_n \in \mathcal{Z}$, and a loss function $\ell : \Theta \times \mathcal{Z} \to \mathbb{R}$, the *population risk* (expected loss) for model $\theta$ is defined to be:
$$R(\theta) = \mathbb{E}[\ell(\theta; Z)].$$
ERM minimizes $\frac{1}{n} \sum_{i=1}^{n} \ell(\theta; Z_i)$ (empirical risk). However, expectation is only one of the many risk measures. For example, value-at-risk and conditional value-at-risk [26] are popular risk measures for evaluating risks of portfolios of financial instruments. CPT defines another way of measuring risk, which aligns with human's preferences. We want to study if minimizing a human-aligned risk will give us ML models that have properties other than a low population risk.

# 3 Human Risk Minimization

**Definition 2.** *Given a real-valued random variable $Z \in \mathcal{Z}$, a loss function $\ell : \Theta \times \mathcal{Z} \to \mathbb{R}$ and a CPT probability weighing function $w \in \overline{\mathcal{W}}_{CPT}$, the human risk is defined to be*

$$R_H(\theta; w) \stackrel{\text{def}}{=} \mathbb{E}[\ell(\theta; Z)w'(F(\ell(\theta; Z)))], \tag{4}$$

*where $F(\ell(\theta; Z))$ is the cumulative distribution function of the loss.*

Comparing Equation 4 with the CPT-weighted rank-dependent utility in Equation 2, we see that $R_H(\theta) = U_{\text{CPT-RD}}(\ell(\theta; Z))$. Given $n$ i.i.d. samples $Z_1, \ldots, Z_n \in \mathcal{Z}$, we define empirical human risk minimization (EHRM) as

$$\theta^* = \arg\min_{\theta} \frac{1}{n} \sum_{i=1}^{n} \ell(\theta; Z_i)w'(F_n(\ell(\theta; Z_i))), \tag{5}$$

where $F_n$ is the empirical CDF of the loss.

**Optimization.** When $\ell$ is differentiable, we use the following iterative update rule to minimize empirical human risk:

$$\theta^{t+1} = \theta^t - \frac{\eta_t}{n} \sum_{i=1}^{n} w_i^t \nabla_\theta \ell(\theta^t; Z_i) \ \text{ for all } \ t \in \{0, \cdots, T-1\},$$

where $w_i^t = w'(F_n(\ell(\theta^t; Z_i)))$ and $\eta_t$ is the learning rate. Note that this heuristic approach relies crucially on the assumption that minor perturbations in $\theta$, don't change $w'(F_n(\ell(\theta; Z))$ drastically. We empirically show that such a heuristic approach performs quite well in practice (See Appendix B). Deriving provably optimal optimization algorithms for EHRM is an interesting open problem.

*Remarks.* In general, there are two levels of decision making in supervised learning: model selection when training a model, and instance prediction when using a model. These two kinds of decisions are very much related. In traditional ERM, a model is selected over others when per-instance predictions are more accurate on average. We explore the consequences of EHRM in both settings: Section 2.1 and 4.2 discuss the model selection consequences of EHRM; Section 5 explores its consequences on per-instance predictions. Different from traditional settings where CPT is considered, supervised learning only evaluates losses. Humans tend to be risk-averse when facing possibilities of large loss. Such a property distinguishes EHRM from ERM. When training machine learning models, surrogate losses are used (e.g., hinge loss is used in replace of $0/1$ loss). Most of the times, such surrogate losses are upper bounds for the original losses. In such cases, the risk-aversion towards possible drastic loss will be carried through when surrogate loss is used instead of the true loss.

To further understand how adding the weights $w'(F(\ell(\theta; Z)))$ to expected loss influences the learned model, we provide the psychological interpretation (Section 4.1) and an analytical illustration of how skewness of the loss distribution may influence choices of people with different risk preferences (Section 4.2) as well as an information weighting view point (Section 4.3) of the probability weighting function.

# 4 Characteristic Properties of Human Risk Minimization

We next review some characteristic properties of human risk minimization, contrasting it with the standard machine learning objective of expected loss minimization. To simplify the notation, we will denote the CDF $F(\ell(\theta; Z))$ of the loss random variable $L$ as $F(\ell)$ in the subsequent analysis.

## 4.1 Diminishing Sensitivity to Probability Changes

Recall from Equation (3) that we work with the following polynomial form of CPT probability weighting function

$$w_{\text{POLY}}(F(x)) = \frac{3 - 3b}{a^2 - a + 1} \left( F(x)^3 - (a+1)F(x)^2 + aF(x) \right) + F(x),$$

where $a \in (0, 1)$ is the fixed point of the function, and $b \in (0, 1)$ controls the curvature. For any event $E$ with probability $P(E) \in (0, 1)$, given a probability change $\Delta$, we define

$$g(P(E)) = \left( w_{\text{POLY}}(P(E) + \Delta; a, b) - w_{\text{POLY}}(P(E); a, b) \right) / \Delta,$$

which is the ratio between the human perceived probability change and the original probability change. Intuitively, $g(P(E))$ represents human's sensitivity to probability changes.

**Lemma 1.** *For any event $E$ with probability $P(E) \in (0, 1)$, $\lim_{\Delta \to 0} g(P(E))$ is a monotonically increasing function of $|P(E) - \frac{a+1}{3}|$.*

The above stated result can be seen as a quantitative evidence of how CPT probability weighting function captures humans' diminishing sensitivity, which has been long-studied in behavioral economics [28]. Humans are sensitive to probability changes of extreme events. Such sensitivity diminishes as the events become less extreme. When using CPT probability weighting function to weight $F(\ell)$, the event we are considering is $E = \mathbb{I}\{L \le \ell\}$, i.e. if the loss $L$ is less than or equal to a threshold $\ell$. In this case, $P(E) = F(\ell)$. Diminishing sensitivity states that for a given amount of probability change, human's perceived probability change depends on where the probability change happens. The perceived change diminishes as the distance between where it happens and the boundary (impossibility $F(\ell) = 0$ and certainty $F(\ell) = 1$) becomes smaller. The probability changes that happen close to the boundary will be up-weighted while the changes in between will be down-weighted. As shown in Lemma 1, for $w_{\text{POLY}}$, the sensitivity of the probability change $\Delta$ diminishes as $P(E)$ moves away from 0 (impossibility) and 1 (certainty).

### 4.2 Responsiveness to Skewness of the Loss Distribution

Since the inverse S-shaped probability weighting function exaggerates small probabilities of both good and bad extreme outcomes, intuitively, its overall impact on evaluating a model depends on higher-order moments of the loss distribution. We highlight this phenomena by considering a family of Bernoulli distributions with same mean and variance. In particular, consider the family of models $\{\mathcal{M}_\theta \mid \theta \in [0, 1]\}$ whose losses $\{\ell(\theta) \mid \theta \in [0, 1]\}$ are parameterized by $\theta$. For all $\theta \in [0, 1]$, suppose that $\ell(\theta)$ follows a Bernoulli distribution [15, 20]:

$$P\left(\ell(\theta) = 1 - \left(\frac{1-\theta}{\theta}\right)^{1/2}\right) = \theta, P\left(\ell(\theta) = 1 + \left(\frac{\theta}{1-\theta}\right)^{1/2}\right) = 1 - \theta.$$

In the above setup, the mean and variance of the losses are independent of $\theta$. In particular, we have that

$$\mathbb{E}[\ell(\theta)] = 1 \text{ and } \text{Var}(\ell(\theta)) = 1 \text{ for all } \theta \in [0, 1].$$

Hence, in this setting empirical risk minimization will treat all the models equally. However, the third central moment (skewness) of $\ell(\theta)$ is given by

$$\text{Skewness}(\ell(\theta)) = \frac{2\theta - 1}{\sqrt{\theta(1-\theta)}}.$$

Observe that $\text{Skewness}(\ell(\theta))$ is a monotonically increasing function of $\theta$, with $\text{Skewness}(\ell(\theta)) = 0$ for $\theta = \frac{1}{2}$. Hence, $\theta < 0.5$ corresponds to models with negatively skewed loss distributions, while $\theta > 0.5$ corresponds to models with positively skewed loss distributions. Then, in this setting, we have the following result:

**Lemma 2.** *Consider the human risk objective in Equation (4) instantiated with $w_{POLY}$ having fixed point $a = \frac{1}{2}$. Then, we have the following:*

1. *For $\theta < 0.5$, $R_H(\theta; w_{POLY}(\,\cdot\,; a, b))$ is a monotonically increasing function of $b$.*

2. *For $\theta > 0.5$, $R_H(\theta; w_{POLY}(\,\cdot\,; a, b))$ is a monotonically decreasing function of $b$.*

*Remarks.* The above result shows that for models with negatively skewed loss distributions, their expected loss is higher than any human risk, while the opposite is true for positively skewed loss distributions. While empirical risk minimization will treat all the models equally, human risk minimization will distinguish the models through higher-order moments of the loss distribution.

### 4.3 Weighting by Information Content

The information content or *"surprisal"* of an event is the amount of information gained when the event is observed and is defined as follows.

**Definition 3.** *The information content of an event $E$ with probability $P(E)$ is defined as*

$$I(E) \overset{\text{def}}{=} -\ln[P(E)],$$

*where $e$ is used as the base of the logarithm.*

Note that the above definition captures the intuition that the observation of a rare event provides more information than a common one. In our setting, the rare event corresponds to the event that the loss $L$ takes extreme values.

Next, we construct a special weighting function $w_{\text{IT}}(\cdot)$ using the information content of the events $E_1 = \mathbb{I}\{L \leq \ell\}$ and $E_2 = \mathbb{I}\{L > \ell\}$. Observe that the information content of the events $E_1$ and $E_2$ is given by $-\ln F(\ell)$ and $-\ln(1 - F(\ell))$ respectively. Moreover, it is easy to see that as $\ell$ gets smaller, the information content of the left tail event $E_1$ increases; and as $\ell$ gets larger, the information content of the right tail event $E_2$ increases.

Then, we can use the information content of $E_1$ and $E_2$ to weight the density of $L$, and define the corresponding weighting function. In particular, the information weighted density is defined to be:

$$w'_{\text{IT}}(F(\ell))f(\ell) = \frac{1}{2}(I(E_1) + I(E_2))f(\ell)$$
$$= -\frac{1}{2}f(\ell)\ln\left(F(\ell) \cdot (1 - F(\ell))\right),$$

and the corresponding **information content weighting function** is given by:

$$w_{\text{IT}}(F(\ell)) = \int_0^1 1 \cdot w'_{\text{IT}}(F(\ell))dF(\ell)$$
$$= \frac{1}{2}\left((1 - F(\ell)) \cdot \ln(1 - F(\ell)) - F(\ell) \cdot \ln(F(\ell))\right) + F(\ell).$$

**Lemma 3.** *The information content weighing function $w_{IT}$ belongs to $\overline{\mathcal{W}}_{CPT}$.*

Interestingly, using $w_{\text{IT}}$ to weight a distribution is of interest in information theory, where it is known as the two-sided information-weighted distribution [6]. Moreover, it is easy to see that $w_{\text{POLY}}(\cdot, a, b)$ with fixed point $a = 1/2$ and curvature $b = \ln 2$ is approximately equal to the third order Taylor approximation of $w_{\text{IT}}$. To the best of our knowledge, this is the first time that information weighting function and CPT probability weighting function have been connected. Uncertainty-aversion in human preferences has also been studied in behavioral economics [3, 4]. In the context of human risk minimization, using $w_{\text{IT}}$, we can define the information-weighted human risk to be

$$R_H(\theta; w_{\text{IT}}) = \mathbb{E}[\ell(\theta; Z)w'_{\text{IT}}(F(\ell(\theta; Z)))].$$

As studied in [6], the information-weighted distribution $w_{\text{IT}}(F(\ell))$ will be heavy-tailed when the CDF $F(\ell) = e^{-\kappa|\ell|}$ for some $\kappa > 0$. Such heavy-tailedness for the loss distribution may cause human risk to be hard to estimate. We believe that deriving statistically and computationally optimal procedures for minimizing $R_H(\theta; w_{\text{IT}})$ is an interesting direction for future work.

## 5 Implications for Performance over Subgroups

Machine learning models are being increasingly deployed to automate a variety of day-to-day tasks. Employers use such models to select job applicants, provide credit scoring and predicting insurance premiums. With such high stakes, ensuring that learned models are non-discriminatory or *fair* with respect to sensitive features such as gender and race is of utmost importance [10, 18, 19, 22]. In this section, we explore the implications of HRM towards subgroup performances. In particular, we know that HRM up-weights possible extreme events, hence, we expect HRM to avoid drastic losses for all subpopulations. We test this hypothesis on both synthetic and real-world datasets and use $w_{\text{POLY}}(\,\cdot\,; a, b)$ specified in Equation 3. We have chosen $a = .5$ so that for all $b \in (0, 1)$, $w'_{\text{POLY}}(F(\ell))$ is symmetric about the line $F(\ell) = a$.

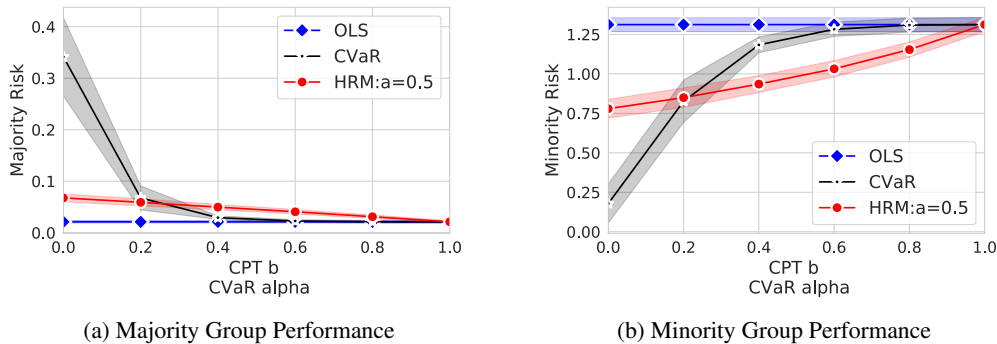

| (a) Majority Group Performance | (b) Minority Group Performance |

Figure 2: Majority and minority performance of ERM, EHRM and CVaR on the synthetic dataset. Note that when $b = 1$, the CPT probability weighting function is the identity function. Hence, EHRM is the same as ERM. 2000 training and 20000 testing data points are used in the experiment. Solid lines and shaded area represent the means and one standard derivations of the risks.

## 5.1 Synthetic Experiment

**Setup.** In this experiment, we create a synthetic regression task to test the performance of EHRM on the minority subgroup. We follow the setup of [9] and draw our covariates (features) from an isotropic Gaussian $X \sim \mathcal{N}(0, \mathbf{I}_5)$ in $\mathbb{R}^5$. The noise distribution is fixed as $\epsilon \sim \mathcal{N}(0, .01)$. We draw our response variable $Y$ as,

$$Y = \begin{cases} X^\top \theta^* + \epsilon & \text{if } X^{(1)} \leq 1.645 \\ X^\top \theta^* + X^{(1)} + \epsilon & \text{otherwise} \end{cases}$$

where $\theta^* = [1, 1, 1, 1, 1]$ and $X^{(1)}$ is the first coordinate of $X$. Observe that since $P\left(X^{(1)} > 1.645\right) = .05$, $\{X \mid X^{(1)} > 1.645\}$ represents our minority subgroup. We fix the squared error $\ell(\theta; (x, y)) = \frac{1}{2}(y - x^T\theta)^2$ as our loss function.

**Results.** Figure 2 plots the risk of minority and majority groups for EHRM and ERM. The empirical risk minimizer is denoted by OLS, the solution of this ordinary least square problem. We see that for different values of $b < 1$, EHRM has a lower minority risk than ERM. Moreover, as $b$ approaches 1, EHRM becomes more similar to ERM. This validates our hypothesis: because the inverse S-shaped probability weighting function inflates small probabilities for extreme losses, drastic losses of the minority group will be exaggerated and human risk minimization trades a low population risk for a better minority performance. Optimization performance of EHRM is shown in Figure 4 (Appendix B). In addition to comparing with ERM, we have also compared EHRM with conditional value-at-risk (CVaR), a risk measure that has been used to measure the worst-case subgroup performance [8, 29]. $\text{CVaR}_\alpha(\ell(\theta; (x, y)))$ is the expectation of the worst $\alpha$ proportion of the losses. As shown in Figure 2, when $\alpha$ is small, CVaR has a lower minority risk than EHRM and ERM, at a cost of a higher majority risk. As $\alpha$ approaches 1, the minority risk increases drastically.

## 5.2 Recidivism Prediction: Similar Subpopulation Performance

**Setup.** We follow the experimental set up in [8]. Using the fairML toolkit version of the COM-PAS recidivism dataset [1], we want to study the performance of EHRM and ERM on different demographic subgroups. With a 90% and 10% train-test split, ERM and EHRM are used to train a logistic regression model with $L_2-$regularization. To study the subgroup performances, we report the misclassification rate of different demographic groups on the test set. In particular, out of the 10 binary features in the dataset, we have chosen 7 of them that have more than 10 samples to group the population. For each chosen (binary) attribute, the dataset can be divided into two subgroups. For EHRM, we have chosen $b$ to be .3 so that the EHRM probability weighting function is close to the median estimate of the CPT probability weighting function for high rank losses in [28]. In practice, $a$ and $b$ are application-dependent and user-dependent.

**Results.** In Figure 3, for each attribute, we report the maximum misclassification rate of the two subgroups at test time. Compared to ERM, EHRM has a higher misclassification rate but a more similar worst case performance across different subgroups. Such an observation aligns with our hypothesis that EHRM avoids extremely bad performances for all demographic groups and hence will sacrifice average performance for similar subpopulation performances.

## 5.3 Gender Classification based on Facial Image: Fairness Metrics Comparison

**Setup.** To study EHRM performance on standard fairness metrics, we use the AI Fairness 360 toolkit [2]. In particular, we use the UTKFace dataset [31] to train a neural network[1] for predicting gender based on facial images (male$= 0$,

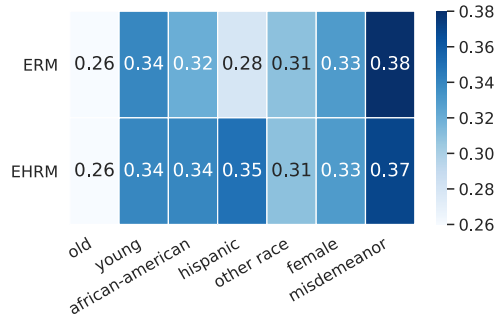

Figure 3: Test misclassification rate of the recidivism prediction task. Each box represents the worst performance among the subpopulations grouped by the attribute listed below. EHRM has a more similar performance across subgroups.

female$= 1$). As suggested by [2], we use race as an indicator to divide the population into two groups $G_1$ (white) and $G_2$ (other race). The fairness metrics we have used include statistical parity difference, disparate impact, equal opportunity difference, average odds difference, Theil index and false negative rate difference.[2] We train the model with ERM and EHRM over 10 random seeds. For EHRM, $b$ is chosen to be .3 for the same reason mentioned in Section 5.2. To minimize empirical human risk, we have used a variant of mini-batch stochastic gradient descent. At each step $t$, $\theta^{t+1} = \theta^t - \eta_t \sum_{i=1}^B w_i^t \nabla_\theta \ell(\theta; Z_i)/B$, where $w_i^t = w'_{\text{POLY}}(F_n(\ell(\theta^t; Z_i)))$, $F_n(\cdot)$ is the empirical CDF of the mini-batch losses, $B$ is the mini-batch size and $\eta_t$ is the learning rate. As shown in Figure 5 (Appendix B), the empirical human risk of the entire training dataset decreases as training proceeds. We have also compared EHRM with a data pre-processing algorithm named reweighing [17] that re-weights the samples so that statistical dependence between the protected attribute and label are mitigated.

**Results.** Table 1 shows the mean and standard deviation of the test time performance of EHRM, and ERM with and without reweighing pre-processing. ERM performs the best in terms of accuracy at test time. However, reweighing with ERM and EHRM does better in terms of the fairness metrics. The empirical result suggests that the human innate risk-aversion towards possibility of extreme losses has promoted similar performances across different subgroups.

Table 1: Mean and standard deviation of accuracy and fairness metrics of models learned by EHRM with $w_{\text{POLY}}(\,\cdot\,; .5, .3)$, and ERM with and without reweighing pre-processing. For each metric, the best performing algorithm is highlighted.

|  | EHRM(.5, .3) | Reweighing [17] | ERM |
|---|---|---|---|
| Accuracy | .8751 ±.0052 | .8767 ±.0067 | **.8767 ±.0060** |
| Stat. Parity Diff. | **-.0825 ±.0220** | -.0875 ±.0212 | -.0881 ±.0208 |
| Disparate Impact | **.8475 ±.0411** | .8396 ±.0390 | .8368±.0386 |
| Equal Opp. Diff. | **-.0440 ±.0261** | -.0518 ±.0253 | -.0502±.0263 |
| Avg. Odds Diff. | **-.0116 ±.0202** | -.0165 ±.0177 | -.0173 ±.0188 |
| Theil Index | .0859 ±.0058 | **.0824 ±.0038** | .0855 ±.0071 |
| FNR Diff. | **.0440 ±.0261** | .0518 ±.0253 | .0502 ±.0263 |

# 6 Conclusion

In this work, we have studied alternatives to empirical risk minimization, and in particular proposed alternate formulations, which are better aligned with human risk measures. We have analyzed several characteristics of human risk minimization such as diminishing sensitivity, model selection based on higher-order moments and information-weighted loss distributions. Further, our empirical analysis has shown that such risk measures have implications for fairness, and in particular trade average performance for similar subgroup performances. Our empirical analysis raises several interesting future directions. Fairness is only one of such desiderata that people start caring about in ML. We would like to study other desiderata that HRM brings. Meanwhile, many risk measures such as conditional value-at-risk [26] can be expressed in a dual form, however, it is not immediately clear if HRM has an equivalent formulation.

**Acknowledgement**   We thank the reviewers for providing thoughtful and constructive feedback for the paper. We thank Hongseok Namkoong for providing the method to optimize conditional value-at-risk. We acknowledge the support of ONR via N000141812861.

## Footnotes

[1]Appendix D consists details of the model configuration.

[2]Appendix C contains definitions of these metrics in terms of $G_1$ and $G_2$.

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
