[Supplementary Material · hrm_nov_8th_supp.pdf]

# A  Proofs

*Proof of Proposition 1.* $w(F(x))$ satisfies: (i) non-decreasing since both $w(\cdot)$ and $F(\cdot)$ are non-decreasing; (ii) $\lim\limits_{x\to+\infty} w(F(x)) = w\left(\lim\limits_{x\to+\infty} F(x)\right) = 1$, $\lim\limits_{x\to-\infty} w(F(x)) = w\left(\lim\limits_{x\to-\infty} F(x)\right) = 0$; and (iii) right continuous since both $w(\cdot)$ and $F(\cdot)$ are continuous. Thus, as shown in Theorem 1.2.2 in [7], $w(F(x))$ is a cumulative distribution function.

*Proof of Lemma 1.* First, notice that $w_{\text{POLY}}(P(E))' = \lim\limits_{\Delta\to0}\big(w_{\text{POLY}}(P(E) + \Delta; a, b) - w_{\text{POLY}}(P(E); a, b)\big)/\Delta$. Denote $P(E)$ to be $y$. Then, $w_{\text{POLY}}(y)' = \frac{3(3-3b)}{a^2-a+1}\left(y^2 - \frac{2(a+1)}{3}y + \frac{a}{3}\right) + 1 = \frac{3(3-3b)}{a^2-a+1}\left((y - \frac{a+1}{3})^2 + (\frac{a}{3} - \frac{(a+1)^2}{9})\right) + 1$. Let $u = |y - \frac{a+1}{3}|$. Then, $w'(y) = \frac{3(3-3b)}{a^2-a+1}\left(u^2 + (\frac{a}{3} - \frac{(a+1)^2}{9})\right) + 1 = f(u)$. $f'(u) = \frac{6(3-3b)}{a^2-a+1}u \geq 0$. Therefore, $f(u)$ is a monotonically increasing function of $u$. Thus, the lemma follows.

*Proof of Lemma 2.* Recall that we are given a family of models $\{\mathcal{M}_\theta \mid \theta \in [0, 1]\}$ whose losses $\{\ell(\theta) \mid \theta \in [0, 1]\}$ are parameterized by $\theta$. For all $\theta \in [0, 1]$, $\ell(\theta)$ follows a Bernoulli distribution:

$$P\left(\ell(\theta) = 1 - \left(\frac{1-\theta}{\theta}\right)^{1/2}\right) = \theta, P\left(\ell(\theta) = 1 + \left(\frac{\theta}{1-\theta}\right)^{1/2}\right) = 1 - \theta.$$

The mean and variance of the losses are the same, i.e. $\mathbb{E}[\ell(\theta)] = 1$ and $\text{Var}(\ell(\theta)) = 1$. Thus, the skewness of $\ell(\theta)$ is

$$\text{Skewness}(\ell(\theta)) = \frac{2\theta - 1}{\sqrt{\theta(1-\theta)}}.$$

Denote $\alpha_\theta = 1 - \left(\frac{1-\theta}{\theta}\right)^{1/2}$, $\beta_\theta = 1 + \left(\frac{\theta}{1-\theta}\right)^{1/2}$ and $\alpha_\theta \leq 1 \leq \beta_\theta$.

$$R_H(\ell(\theta); w) = \alpha_\theta(w(\theta) - w(0)) + \beta_\theta(w(1) - w(\theta)) = \alpha_\theta w(\theta) + \beta_\theta(1 - w(\theta)).^3$$

Then, for different polynomial form of CPT probability weighting function $w_1$ and $w_2$,

$$R_H(\ell(\theta); w_1) - R_H(\ell(\theta); w_2) = (\beta_\theta - \alpha_\theta)(w_2(\theta) - w_1(\theta)).$$

Suppose that the probability weighting functions $w_1, w_2$ have the parametric form suggested by Equation 3 with parameter $a_1, b_1$ and $a_2, b_2$ respectively. If $a_1 = a_2 = \frac{1}{2}$ and $b_1 < b_2$, then $w_1(\theta) > w_2(\theta)$ on $[0, .5)$ and $w_1(\theta) < w_2(\theta)$ on $(.5, 1]$.

- If $0 \leq \theta < .5$, then $R_H(\theta; w_1) < R_H(\theta; w_2)$.
- If $.5 < \theta \leq 1$, then $R_H(\theta; w_2) < R_H(\theta; w_1)$.

*Proof of Lemma 3.* First, notice that $w_{\text{IT}}(0) = 0$ and $w_{\text{IT}}(1) = 1$. The fixed point of $w_{\text{IT}}$ is $\frac{1}{2}$ because $w_{\text{IT}}(1/2) = 1/2$. Since $w'_{\text{IT}}(y) = -1/2\ln(y \cdot (1 - y)) > 0$ for all $x \in [0, 1]$, $w_{\text{IT}}$ is monotonically increasing. Notice that $w''_{\text{IT}}(y) = \frac{2y-1}{2y(1-y)}$. Since $w''_{\text{IT}}(y) < 0$ for all $y \in [0, \frac{1}{2})$ and $w''_{\text{IT}}(y) > 0$ for all $y \in (\frac{1}{2}, 1]$, $w'_{\text{IT}}$ is monotonically decreasing on $[0, \frac{1}{2})$ and monotonically increasing on $(\frac{1}{2}, 1]$. Thus, $w_{\text{IT}} \in \overline{\mathcal{W}}_{\text{CPT}}$.

*Connection between $w_{IT}$ and $w_{POLY}$.* Let $y = F(\ell)$ and $w'_{\text{IT}}(y) = \frac{dw_{\text{IT}}(y)}{dy}$.

$$
\begin{aligned}
w_{\text{POLY}}(y; 1/2, \ln 2) &= 4y^3 - 6y^2 + 3y + \left(-4\ln 2y^3 + 6\ln 2y^2 - 2\ln 2y\right) \\
&= 4y^3 - 6y^2 + 3y + g(y) \\
&\approx 4y^3 - 6y^2 + 3y + g(1/2) + g'(1/2)(y - 1/2) \\
&= 4y^3 - 6y^2 + 3y + (\ln 2y - \ln 2/2) \\
&= w_{\text{IT}}(1/2) + w'_{\text{IT}}(1/2)(y - 1/2) + \frac{w''_{\text{IT}}(1/2)}{2}(y - 1/2)^2 + \frac{w'''_{\text{IT}}(1/2)}{6}(y - 1/2)^2 \\
&\approx w_{\text{IT}}(y).
\end{aligned}
$$

The first approximation is done by the first order Taylor expansion of $g(y)$ around $1/2$ and the second approximation is done through the third order Taylor expansion of $w_{\text{IT}}$ around $1/2$.

## B Optimization of EHRM

Optimally optimizing empirical human-aligned risk is an interesting open question. However, the heuristic approach described in Section 3 performs relatively well in the experiments. Figure 4 and 5 show the empirical human risk (at training time) of experiments in Section 5.1 and 5.3 respectively.

(a) $a = .5, b = .4$

(b) $a = .5, b = .8$

Figure 4: Using a fixed learning rate .05 and optimization method described in Section 3, empirical human risk of the experiments in Section 5.1 converge within 100 iterations.

(a) $a = .5, b = .3$

Figure 5: Using the optimization method described in Section 5.3, empirical human risk of the experiment (Section 5.3) converges in 100 epochs.

## C Fairness Metrics

Denote true positive rate as TPR, false positive rate as FPR, false negative rate as FNG, covariate as $X \in \mathcal{X}$ and label as $Y \in \{0, 1\}$. As suggested by [2], we define the below fairness metrics in terms of the privileged group $G_1 \subseteq \mathcal{X}$ and unprivileged group $G_2 \subseteq \mathcal{X}$:

1. Statistical Parity Difference: $P(Y = 1 | X \in G_2) - P(Y = 1 | X \in G_1)$.

2. Disparate Impact: $\frac{P(Y=1|X \in G_2)}{P(Y=1|X \in G_1)}$.

3. Equal Opportunity Difference: $\text{TPR}(G_2) - \text{TPR}(G_1)$.

4. Average Odds Difference: $\frac{1}{2}(\text{FPR}(G_2) - \text{FPR}(G_1) + (\text{TPR}(G_2) - \text{TPR}(G_1)))$.

5. Theil Index: $\frac{1}{n}\sum_{i=1}^{n} \frac{b_i}{\mu} \ln(\frac{b_i}{\mu})$ where $b_i = \widehat{Y}_i - Y_i + 1$ and $\mu = \frac{1}{n}\sum_{i=1}^{n} b_i$. $\widehat{Y}_i$ is the prediction of $X_i$ and $n$ is the number of samples.

6. False Negative Rate Difference: $\text{FNR}(G_2) - \text{FNR}(G_1)$.

# D    Model Configuration

The model configuration for the gender classification task (Section 5.3) is as follows: 3 convolutional layers (with number of output channels $(6, 16, 16)$ respectively, kernel size $(5, 5, 6)$ respectively and a $2 \times 2$ max-pooling on the outputs of the first layer), followed by two fully connected layers (the first has 120 hidden units and the second is the output layer with 2 output units); all activation functions are ReLU and all convolutional layers use stride 1.

## Footnotes

[3]The loss distribution is discrete in this case. We have used the CPT-weighted rank-dependent utility of a discrete random variable to obtain the human risk [5].