[Reviews · NeurIPS 2019]

Reviewer 1



The paper presents a novel alternative to ERM that is based on an economic theory called CPT which is composed of an inverse S-shaped probability weighting function to transform the CDF so that the small probabilities are inflated and large probabilities are deflated. The only similar works considering a human loss/reward/risk have been studied in Bandits and RL. I am not aware of other literature that studies in the context of supervised learning for classification, although I might be missing something here. The paper is well written and very clear in most arguments it makes. There are little to no typos except ones noted below. Weaknesses: 0. My first concern is the assumption that a human risk measure is gold standard when it comes to fairness. There are many reasons to question this assumption. First, humans are the worst random number generators, e.g. the distribution over random integers from 1 to 10 is highly skewed in the center. Similarly, if humans perceive a higher risk in the tails of a distribution, it doesn't necessarily mean that minimizing such risk makes the model fair. This still needs to be discussed and proven. 1. The paper suggests that using EHRM has fairness implications. These fairness implications are obtained as a side effect of using different hyperparameter setting for the skewness of the human risk distribution. There is no direct relationship between fairness consideration and the risk metric used. 2. In the Introduction, the authors choose to over-sell their work by presenting their work as a "very natural if simple solution to addressing these varied desiderata" where the desiderata include "fairness, safety, and robustness". This is a strong statement but incorrect at the same time. The paper lacks any connection between these objectives and the proposed risk metric. One could try to investigate these connections before claiming to address them. 3. One example of connection would be the definition of Calibration used in, for example, Kleinberg et al. and connect it to a human calibration measure and derive a Human risk objective from there as well. It is a straightforward application but the work lacks that. 4. There are no comparison baselines even when applying to a fairness problem which has a number of available software to get good results. Agarwal 2018: "A Reductions Approach to Fair Classification" is seemingly relevant as it reduces fairness in classification to cost-sensitive learning. In this case, the weighting is done on the basis of the loss and not the group identities or class values, but it may be the reason why there is a slight improvement in fairness outcomes. Since the EHRM weights minorities higher, it might be correlated to the weights under a fair classification reduction and hence giving you slight improvements in fairness metrics. 5. There were a few typos and some other mistakes: - doomed -> deemed (Line50) - Line 74: Remove hence. The last line doesn't imply this sentence. It seems independent.

Reviewer 2



Thank you for your response. I would love to see this paper published, but feel that there are still issues that should be addressed: - More convincing experimental results (mostly in terms of effect size) - Better justification as to why fairness is expected to improve under HRM (since, as noted in the response, all subgroups should be effected) - Decoupling the part risk aversion plays in model selection vs. prediction And while my score remains unchanged, I truly hope that a revised and improved version of this paper will be published in the near future. -------------------------------------------------------------------------------------------------------- The paper explores a conceptually intriguing question - what if predictive machines were trained with imbued human biases (e.g., loss aversion)? The authors propose that this would trade off accuracy for other measures of interest such as fairness, which they aim to show empirically. Overall the paper makes an interesting point and offers a novel perspective, and the paper generally reads well. There are however several issues that I'm hoping the authors can elaborate on. 1. I'm having trouble untangling what aspect of the machine learning process is drawn the analogy to human decision making. In some parts of the paper, human preferences are linked to model selection (as in the example in Sec. 2.1, and in general in its relation to the ERM setup). However, the paper is motivated by the increasing use of machines in real-world decision making, which relates not to model selection but to per-instance decisions (made on the basis of predictions). 2. In Def. 1, F is portrayed as 'given'. Is it indeed a given object, or is it the cdf induced by the data distribution (through \ell) (which makes more sense here)? 3. How do the authors propose to optimize the EHRM in Eq. 5? Specifically, how is F and/or F_n modeled and parameterized? Is it computationally efficient to train? In general, F can be a complex function of \theta, and possibly non-differentiable (e.g., if it bins values of \ell). 4. It is not clear which parts of Sec. 4 are novel and which are not. 5. The link between human-aligned risk and fairness is presented in line 176: "... we expect minimizers of HRM to avoid drastic losses for minority groups." I understand why HRM avoids drastic losses, but why would these necessarily account for those of minority groups? This is an important argument that the authors should clarify. 6. The empirical section (which is stated as being a main contribution) is somewhat discouraging, specifically: - he effects seem small and noisy (especially in the figures) - the table is missing many of the stated model configurations - the table lacks significance tests (that should correct for multiple hypotheses, of which there are fairly many) - there is no comparison to other fairness-promoting baselines - Sec. 5.3 considers only one configuration (without and explanation) - feature weights is not a robust (or meaningful) criterion for comparing models

Reviewer 3



Orginality: Moderate to high The idea of using better, more human-aligned risks is a nice idea that extends recent interest in having more risk-averse losses and training procedures. Approaching the problem through the lens of behavioral economics is different from many of the machine learning based ideas. Quality: Moderate. The evaluation and comparisons to prior work are sound (with some minor omissions, like `Fairness Risk Measures' by Williamson which cover similar ideas about risk measures and fairness). The paper has minor conceptual issues that prevent it from being a great paper. The main problem is that CPT is justified relatively briefly in section 2, and then a large part of the remaining paper is about the analysis of CPT. While CPT is clearly a reasonable framework for behavioral economics, it is not clear that the (surrogate) losses used in machine learning are at all similar to monetary losses and gains. In this case, should we still assume that CPT is the right way to weight losses? What is missing in this paper is the 'human' in human risk minimization - quantifying whether losses used in ML + CPT matches human utility would go a long way to convincing the reader. Additionally, many of the goals of CPT can be achieved by simply using more risk averse losses like CVaR. Is the upweighting of tail gains necessary? I would think that this can in fact result in unfairness when one tries to optimize tail gains at the expense of losses. It might be worthwhile to include something like CVaR in your experiments as well. Clarity: Moderate to high. Section 2.1 that covers CPT was clear, and motivated the reasons for using CPT. I do think Section 4 is a bit low level, and cutting some material there to show some examples of how CPT would work on toy examples would be helpful. I did not follow the phrasing of the 3 bullet points from lines 126 to 128. Typo (?) at line 50 ``doomed larger than gains''? Significance: Moderate to high Despite some minor complaints about whether CPT truly matches human risk, I think the ideas and approach are new, and the writing is fairly good at making the case for CPT as a way to adjust losses. The paper may serve to interest other researchers in more human-aligned loss measures as well.

[Author Response · NeurIPS 2019]

We thank the reviewers for their kind comments, and for their consensus view that our approach of porting decision theory backed by behavior economics into classical ML is a promising research direction. We are also thankful for the reviewers' concrete suggestions on improving the draft, which we will incorporate in the final version of our work. Based on the requests of the reviewers, we have added an additional fairness baseline to compare with EHRM, using the reweighting approach in *Data preprocessing techniques for classification without discrimination, Kamiran et al.* When compared with EHRM$(.4, .7)$ from the main paper, we see that EHRM performs favorably across a variety of metrics.

**AR1:** *Human-aligned risk and fairness.* We agree with the reviewer that human risk measures are not necessarily fair. However, the innate loss aversion provided by CPT, ensures that HRM-learned models always avoid drastic losses, and consequently ensure that all subgroups do not suffer from huge losses. Whether this will lead to a "fairer" model is what we intended to explore with the experiments. In future work, we will theoretically study the connections between HRM and fairness, and combining CPT and Calibration (Kleinberg et. al. [2016]) to derive human-calibrated risk measures.

*Connection to existing fairness literature.* We thank the reviewer for pointing out the work of Agarwal et. al. [2018]. In contrast to EHRM, the authors' method requires access to an explicit set of protected attributes during training. Nevertheless, it is certainly an interesting question to see how the weights in EHRM are related to the cost-weighted framework of Agarwal et. al. We would also like to point out that fairness is one of several significant facets of our paper. Our primary goal is to introduce CPT inspired risk measures and study the consequences of its use within ML.

**AR2:** *Machine learning and human decision making.* As pointed out by the reviewer, there are two levels of decision making in ML: model selection when training a model, and instance prediction when using a model. These two kinds of decisions are very much related. In traditional ERM, a model is selected over others when per-instance predictions are more accurate on average. Our work explores the consequences of HRM in both settings: Sec. 2.1 and Sec. 4.3 discuss the model selection consequences of HRM; Sec. 5 explores its consequences on per-instance predictions.

Table 1: $a = .4, b = .7$ ensures EHRM weighting function to be close to the median estimate of the CPT weighting function given in Kahneman et al. [1992].

| EHRM(.4, .7) | Kamiran et al. |
|---|---|
| .8766, .0057 | **.8767 ±.0067** |
| **-.0831 ±.0158** | -.0875 ±.0212 |
| **.8453 ±.0293** | .8396 ±.0390 |
| **-.0422 ±.0157** | -.0518 ±.0253 |
| **-.0120 ±.0135** | -.0165 ±.0177 |
| .0861 ±.0034 | **.0824 ±.0038** |
| **.0422 ±.0157** | .0518 ±.0253 |

*Optimizing EHRM.* We use the following iterative optimization procedure: $\theta^{t+1} = \theta^t - \eta \sum_{i=1}^{n} w_i \nabla_\theta \ell(\theta; z_i)$, where $w_i$ are the weights obtained by reweighting the empirical risk using the weighted CDF given by CPT. Note that this approach is still a heuristic, and relies crucially on the assumption that minor perturbations in $\theta$, don't change $w(F_n(\ell(\theta; z)))$. Deriving provably optimal optimization algorithms for EHRM is an interesting open problem.

*Novelty of Section 4.* We would like to point out that (a) information weighted densities have been studied in information theory (Oliveira et. al. [2016]) and (b) effect of the probability weighting function on skewness of a distribution has been studied in finance in the context of portfolio allocations. However, we believe that our observations are novel, as (a) has not been studied in the context of CPT, (b) has not been studied on model selection, and these properties have certainly not been discussed in a unified way in the ML community. We will clarify this further in future versions.

*Defn. 1. and L176.* 1) The word "given" is inappropriate. $F(\ell)$ is the CDF induced by the data distribution. 2) HRM avoids drastic losses for all subgroups, including minority groups. The first paragraph of AR1 contains more details.

*Experiments.* We refer the reviewer to Appendix B (Fig. 5) in the supplementary material for separate plots of majority and minority performance of EHRM and ERM. We will add further figures of FNR with more fine-grained settings of $a, b$. The model configuration for the gender classification task is as follows: 3 convolutional layers (with number of output channels $(6, 16, 16)$ respectively, kernel size $(5, 5, 6)$ respectively and a $2 \times 2$ max-pooling on the outputs of the first layer), followed by two fully connected layers (the first has 120 hidden units and the second is the output layer with 2 output units); all activation functions are ReLU and all convolutional layers use stride 1. We thank the reviewer for bringing up the point of multiple hypothesis testing. We will correct our confidence intervals for this. As noted in Table 1 in the rebuttal, even a single fixed setting of EHRM compares favorably to existing fairness baselines.

**AR3:** *L126-128.* Diminishing sensitivity discusses how humans are less sensitive to certain changes of probability. Since $F(\ell) = 0$ and $F(\ell) = 1$ are "boundary" events (L124-L125), the inverse S-shaped probability weighting function implies that humans are more sensitive to changes of $F(\ell)$ when $F(\ell)$ is closer to 0 or 1 than when $F(\ell) = .3$.

*Related Work.* We thank the reviewer for pointing out *Fairness Risk Measures, Williamson & Menon* and for their suggestion to explore other risk measures such as CVaR. We will include a discussion of the paper in the related works section, and add experiments comparing EHRM and other risk measures.

*Why CPT for surrogate loss minimization?* It is the case that surrogate losses in ML are different from monetary gains and losses studied in behavioral economics. In particular, in ML, risk minimization only considers losses. However, what CPT captures is the characteristics of human's risk preferences when they make decisions under uncertainty. When applying CPT to surrogate loss minimization, we are assuming that an ideal ML model shares similar risk preferences as humans, e.g. avoiding drastic losses. We will make the connection clearer in the final version of the paper. As the reviewer has suggested, quantifying the alignment of HRM and human utility is a promising research direction.

[Meta-Review · NeurIPS 2019]

This paper is well written and well-motivated and makes the following interesting contributions: 1. proposing human-aligned risk measures suitable for ML by constructing the risk measures using cumulative prospect theory -- a novel and interesting idea 2. Establish a connection between the choice of risk estimator and properties (specifically fairness) of the learned predictor. However, based on reviewer feedback, there are also certain aspects/weaknesses that need to be addressed: 1. Multiple reviewers pointed out that it is not entirely cleary why fairness is expected to improve under human risk measure (proposed by this paper). The authors need to provide a clear justification for this. 2. It is also unclear how and why Cumulative Prospect Theory (CPT) matches human risk and why it should be applied to surrogate losses. This should be clearly justified in the paper. 2. Evaluation of this paper seems somewhat weak as pointed out by multiple reviewers: No comparison baselines to fairness problem, feature weights is not a meaningful metric for comparing models, the effects in figures seem quite small and noisy, evidence that CPT matches human utility is missing. We would strongly encourage the authors to address these issues. All in all, this is a borderline paper with some important and interesting contributions, but lacking in experimental evaluation and justification of certain assumptions.